# Three-dimensional imaging through scattering media based on confocal diffuse tomography

David B. Lindell [1✉] & Gordon Wetzstein [1✉]

Optical imaging techniques, such as light detection and ranging (LiDAR), are essential tools in remote sensing, robotic vision, and autonomous driving. However, the presence of scattering places fundamental limits on our ability to image through fog, rain, dust, or the atmosphere. Conventional approaches for imaging through scattering media operate at microscopic scales or require a priori knowledge of the target location for 3D imaging. We introduce a technique that co-designs single-photon avalanche diodes, ultra-fast pulsed lasers, and a new inverse method to capture 3D shape through scattering media. We demonstrate acquisition of shape and position for objects hidden behind a thick diffuser (≈6 transport mean free paths) at macroscopic scales. Our technique, confocal diffuse tomography, may be of considerable value to the aforementioned applications.

[1] Department of Electrical Engineering, Stanford University, 350 Jane Stanford Way, Stanford, CA 94305, USA. ✉email: lindell@stanford.edu; gordon.wetzstein@stanford.edu

Scattering is a physical process that places fundamental limits on all optical imaging systems. For example, light detection and ranging (LiDAR) systems are crucial for automotive, underwater, and aerial vehicles to sense and understand their surrounding 3D environment. Yet, current LiDAR systems fail in adverse conditions where clouds, fog, dust, rain, or murky water induce scattering. This limitation is a critical roadblock for 3D sensing and navigation systems, hindering robust and safe operation. Similar challenges arise in other macroscopic applications relating to remote sensing or astronomy, where an atmospheric scattering layer hinders measurement capture. In microscopic applications, such as biomedical imaging and neuroimaging[1], scattering complicates imaging through tissue or into the brain, and is an obstacle to high-resolution in vivo imaging[2]. Robust, efficient imaging through strongly scattering media in any of these applications is a challenge because it generally requires solving an inverse problem that is highly ill-posed.

Several different approaches have been proposed to address the challenging problem of imaging through and within scattering media. The various techniques can be broadly classified as relying on ballistic photons, interference of light, or being based on diffuse optical tomography. Ballistic photons travel on a direct path through a medium without scattering and can be isolated using time-gating[3,4], coherence-gating[5,6], or coherent probing and detection of a target at different illumination angles[7]. By filtering out scattered photons, the effects of the scattering media can effectively be ignored. While detecting ballistic photons is possible in scattering regimes where the propagation distance is small (e.g., optical coherence tomography[8]), ballistic imaging becomes impractical for greater propagation distances or more highly scattering media because the number of unscattered photons rapidly approaches zero. Moreover, 3D ballistic imaging typically requires a priori knowledge of the target position in order to calibrate the gating mechanism. Alternatively, methods based on interference of light exploit information in the speckle pattern created by the scattered wavefront to recover an image[9–11]; however, these techniques rely on the memory effect, which holds only for a limited angular field of view, making them most suited to microscopic scales. Other interference-based techniques use wavefront shaping to focus light through or within scattering media, but often require invasive access to both sides of the scattering media[12]. Guidestar methods[13] similarly use wavefront shaping, typically relying on fluorescence[14–16] or photoacoustic modulation[17–19] to achieve a sharp focus. Finally, another class of methods reconstructs objects by explicitly modeling and inverting scattering of light. For example, non-line-of-sight imaging techniques invert scattering off of a surface or through a thin layer[20–26], but do not account for diffusive scattering. Diffuse optical tomography (DOT) reconstructs objects within thick scattering media by modeling the diffusion of light from illumination sources to detectors placed around the scattering volume[27,28]. While conventional CMOS detectors have been used for DOT[29,30], time-resolved detection[2,31–36] is promising because it enables direct measurement of the path lengths of scattered photons.

In all cases, techniques for imaging through or within scattering media operate in a tradeoff space: as the depth of the scattering media increases, resolution degrades. So while ballistic imaging and interference-based techniques can achieve micron-scale resolution at microscopic scales[37], for highly scattering media at large scales the resolution worsens and key assumptions fail. For example, the number of ballistic photons drops off, the memory effect no longer holds, and coherent imaging requires long reference arms or becomes a challenge due to

large bandwidth requirements[38]. DOT operates without a requirement for isolating ballistic photons or exploiting interference of light. As such, it is one of the most promising directions for capturing objects obscured by highly scattering media at meter-sized scales or greater with centimeter-scale resolution. Still, current techniques for DOT are often invasive, requiring access to both sides of the scattering media[35,39], limited to 2D reconstruction, or they require computationally expensive iterative inversion procedures with generally limited reconstruction quality[40].

Here, we introduce a technique for noninvasive 3D imaging through scattering media: confocal diffuse tomography (CDT). We apply this technique to a complex and challenging macroscopic imaging regime, modeling and inverting the scattering of photons that travel through a thick diffuser (≈6 transport mean free paths), propagate through free space to a hidden object, and scatter back again through the diffuser. Our insight is that a hardware design specifically patterned after confocal scanning systems (such as commercial LiDARs), combining emerging single-photon-sensitive, picosecond-accurate detectors, and newly developed signal processing transforms, allows for an efficient approximate solution to this challenging inverse problem. By explicitly modeling and inverting scattering processes, CDT incorporates scattered photons into the reconstruction procedure, enabling imaging in regimes where ballistic imaging is too photon inefficient to be effective. CDT enables noninvasive 3D imaging through thick scattering media, a problem which requires modeling and inverting diffusive scattering and free-space propagation of light to a hidden object and back. The approach operates with low computational complexity at relatively long range for large, meter-sized imaging volumes.

## Results

**Experimental setup.** In our experiments, measurements are captured by illuminating points on the surface of the scattering medium using short (≈35 ps) pulses of light from a laser (see Fig. 1). The pulsed laser shares an optical path with a single-pixel, single-photon avalanche diode (SPAD), which is focused on the illuminated point and detects the returning photons (see Supplementary Fig. 1). The SPAD is time gated to prevent saturation of the detector from the direct return of photons from the surface of the scattering medium, preserving sensitivity and bandwidth for photons arriving later in time from the hidden object. A pair of scanning mirrors controlled by a two-axis galvanometer scan the laser and SPAD onto a grid of 32 by 32 points across a roughly 60 by 60 cm area on the scattering medium.

The pulsed laser source has a wavelength of 532 nm and is configured for a pulse repetition rate of 10 MHz with 400 mW average power. For the scattering medium, we use a 2.54-cm thick slab of polyurethane foam. We estimate the scattering properties of the foam by measuring the temporal scattering response and fitting the parameters using a nonlinear regression (see Methods and Supplementary Note 1). The estimated value of the absorption coefficient ($\mu_a$) is $5.26 \times 10^{-3} \pm 5.5 \times 10^{-5} \mathrm{cm}^{-1}$ and the reduced scattering coefficient ($\mu_s'$) is $2.62 \pm 0.43 \mathrm{cm}^{-1}$. Here, the confidence intervals indicate possible variation in the fitted parameters given uncertainty in the modeling coefficients (see Supplementary Figs. 2 and 3). Thus the length of one transport mean free path is ~3.8 ± 0.6 mm, which is several times smaller than the total thickness of the slab, allowing us to approximate the propagation of light through the foam using diffusion.

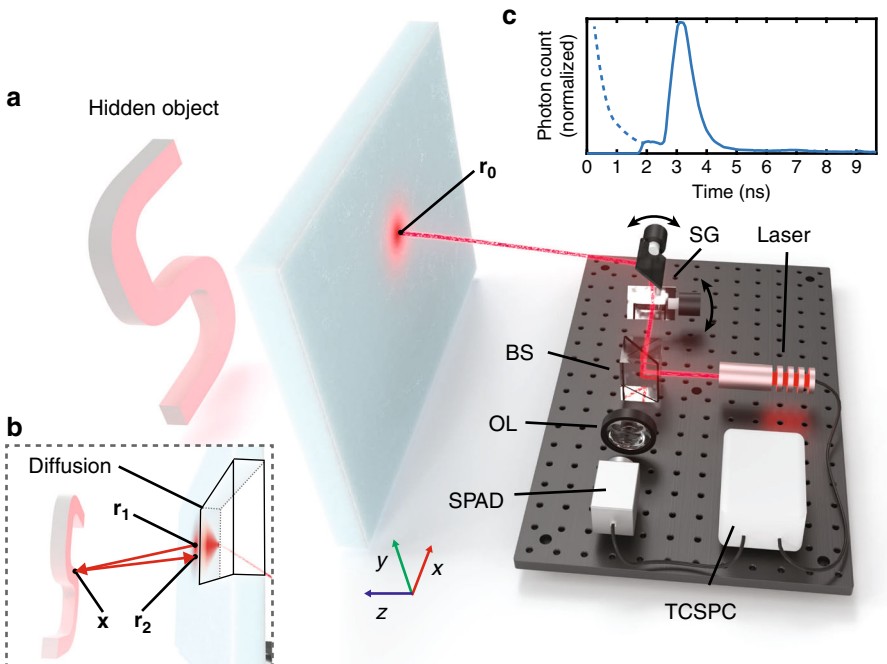

**Fig. 1 Schematic of 3D imaging through scattering media. a** A pulsed laser and time-resolved single-photon detector raster-scan the surface of the scattering medium. **b** Light diffuses through the medium, is back-reflected by the hidden object, and diffuses back through the medium to the detector. **c** Returning photons from the hidden object are captured by the detector over time, with earlier arriving photons being gated out (dashed line). SG scanning galvanometer, BS beamsplitter, OL objective lens, SPAD single-photon avalanche diode, TCSPC time-correlated single-photon counter.

**Image formation model**. To model light transport through the scattering medium, we solve the diffusion equation for the slab geometry of our setup. In this geometry, the physical interface between the scattering medium and the surrounding environment imposes boundary conditions that must be considered. A common approximation is to use an extrapolated boundary condition where the diffusive intensity is assumed to be zero at a flat surface located some extrapolation distance, $z_e$, away from either side of the slab. In other words, for a slab of thickness $z_d$, this condition states that the diffusive intensity is zero at $z = -z_e$ and $z = z_d + z_e$. As we detail in Supplementary Note 2, the value of $z_e$ depends on the amount of internal reflection of diffusive intensity due to the refractive index mismatch at the medium-air interface[41]. To simplify the solution, we further assume that incident photons from a collimated beam of light are initially scattered isotropically at a distance $z_0 = 1/\mu'_s$ into the scattering medium[41–43].

The solution of the diffusion equation satisfies the extrapolated boundary condition by placing a positive and negative (dipole) source about $z = -z_e$ such that the total diffusive intensity at the extrapolation distance is zero. However, a single dipole source does not satisfy the boundary condition at $z = z_d + z_e$. Instead, an infinite number of dipole sources is required, where the dipole of the near interface ($z = -z_e$) is mirrored about the far interface, which is then mirrored about the near interface, and so on, as illustrated in Supplementary Fig. 4. The positions of these positive and negative sources are[44]

$$
\begin{aligned}
z_{+,i} &= 2i(z_d + 2z_e) + z_0 \\
z_{-,i} &= 2i(z_d + 2z_e) - 2z_e - z_0 \; . \\
i &= 0, \pm 1, \pm 2, \ldots
\end{aligned}
\tag{1}
$$

The resulting solution to the diffusion equation is[41,44]

$$
\begin{aligned}
\phi(t, \mathbf{r_0}, \mathbf{r_1}) = & \frac{1}{2(4\pi Dc)^{3/2} \, t^{5/2}} \\
& \exp\left( -\mu_a ct - \frac{\left(r_{1,x} - r_{0,x}\right)^2 + \left(r_{1,y} - r_{0,y}\right)^2}{4Dct} \right) \\
& \cdot \sum_{i=-\infty}^{\infty} \left[ (z_d - z_{+,i}) \exp\left( -\frac{\left(z_d - z_{+,i}\right)^2}{4Dct} \right) \right. \\
& \left. - (z_d - z_{-,i}) \exp\left( -\frac{\left(z_d - z_{-,i}\right)^2}{4Dct} \right) \right],
\end{aligned}
\tag{2}
$$

where $\phi$ is the power transmitted through the slab per unit area, $\mathbf{r_0} \in \Omega_0 = \{(r_{0,x}, r_{0,y}, r_{0,z}) \in \mathbb{R} \times \mathbb{R} \times \mathbb{R} \mid r_{0,z} = 0\}$ is the position illuminated by the laser and imaged by the detector, and $\mathbf{r_1} \in \Omega_{z_d} = \{(r_{1,x}, r_{1,y}, r_{1,z}) \in \mathbb{R} \times \mathbb{R} \times \mathbb{R} \mid r_{1,z} = z_d\}$ is a spatial position on the far side of the scattering medium (see Fig. 1). We also have that $c$ and $t$ are the speed of light within the medium and time, respectively, and $D$ is the diffusion coefficient, given by $D = (3(\mu_a + \mu'_s))^{-1}$. Generally, truncating the solution to 7 dipole pairs (i.e., $i = 0, \pm 1, \pm 2, \pm 3$) is sufficient to reduce the error to a negligible value[44].

The complete measurement model, consisting of diffusion of light through the scattering medium, free-space propagation to and from the hidden object, and diffusion back through the

scattering medium is given as

$$\tau(t, \mathbf{r_0}) = \int_{\Omega_{z_d}} \int_0^\infty \left[ \underbrace{\int_{\Omega_{z_d}} \int_0^\infty \phi(t'' - t', \mathbf{r_0}, \mathbf{r_1})}_{\mathbf{r_0} \to \mathbf{r_1}} \right.$$

$$\underbrace{\left[ \underbrace{\int_\Psi f(\mathbf{x}, \mathbf{r_1}) f(\mathbf{x}, \mathbf{r_2}) \delta(ct' - \| \mathbf{x} - \mathbf{r_1} \| - \| \mathbf{x} - \mathbf{r_2} \|) d\mathbf{x}}_{I(t', \mathbf{r_1}, \mathbf{r_2}):\mathbf{r_1} \to \mathbf{x} \to \mathbf{r_2}} \right] dt' d\mathbf{r_1}}{}$$

$$\left. \underbrace{\phi(t - t'', \mathbf{r_0}, \mathbf{r_2}) dt'' d\mathbf{r_2}}_{\mathbf{r_2} \to \mathbf{r_0}} \right].$$

(3)

Here, the measurements $\tau(t, \mathbf{r_0})$ are described by integrals over three propagation operations (see Fig. 1): (1) the diffusion of light through the scattering medium from point $\mathbf{r_0}$ to $\mathbf{r_1}$ as modeled using Eq. (2), (2) the free-space propagation of light from $\mathbf{r_1}$ to a point $\mathbf{x}$ on the hidden object and back to another point $\mathbf{r_2}$, and (3) the diffusion of light back through the scattering medium from $\mathbf{r_2}$ to $\mathbf{r_0}$. The free-space propagation operator, $I(t', \mathbf{r_1}, \mathbf{r_2})$, is composed of a function, $f$, which describes the light throughput from a point on the scattering medium to a point on the hidden object and incorporates the bidirectional scattering distribution function (BSDF), as well as albedo, visibility, and inverse-square falloff factors[45]. A delta function, $\delta$, relates distance and propagation time, and integration is performed over time and the hidden volume $\mathbf{x} \in \Psi = \{(x, y, z) \in \mathbb{R} \times \mathbb{R} \times \mathbb{R} \mid z \geq z_d\}$.

The measurements can be modeled using Eq. (3); however, inverting this model directly to recover the hidden object is computationally infeasible. The computational complexity is driven by the requirement of convolving the time-resolved transmittance of Eq. (2) with $I(t, \mathbf{r_1}, \mathbf{r_2})$ of Eq. (3) for all light paths from all points $\mathbf{r_1}$ to all points $\mathbf{r_2}$. We introduce an efficient approximation to this model, which takes advantage of our confocal acquisition procedure, where the illumination source and detector share an optical path, and measurements are captured by illuminating and imaging a grid of points on the surface of the scattering medium.

The confocal measurements capture light paths which originate and end at a single illuminated and imaged point on the scattering medium. As light diffuses through the scattering medium, it illuminates a patch on the far side of the scattering medium whose lateral extent is small relative to the axial distance to the hidden object. Likewise, backscattered light incident on that same small patch diffuses back to the detector. We therefore approximate the free-space propagation operator $I$ by modeling only paths that travel from an illuminated point $\mathbf{r_1}$, to the hidden object, and back to the same point. In other words, we make the approximation $\mathbf{r_1} \approx \mathbf{r_2}$. This approximation results in a simplified convolutional image formation model (see Supplementary Note 2, Supplementary Figs. 5 and 6)

$$\hat{\tau}(t, \mathbf{r_0}) = \phi(t, \mathbf{r_0}, \mathbf{r_1}) * \phi(t, \mathbf{r_0}, \mathbf{r_1}) * I(t, \mathbf{r_1}, \mathbf{r_1})$$
$$= \bar{\phi} * I,$$

(4)

where $\hat{\tau}$ is the approximated measurement and $\bar{\phi}$ is a convolutional kernel used to model diffusion through the scattering medium and back.

The continuous convolution operator $\bar{\phi}$ and the continuous free-space propagation operator $I$ are implemented with discrete matrix operations in practice. We denote the discrete diffusion operator as the convolution matrix (or its equivalent matrix-free operation), $\bar{\Phi}$. Then, let $\mathbf{A}$ be the matrix that describes free-space propagation to the hidden object and back, and let $\rho$ represent the

hidden object albedo. The full discretized image formation model is then given as $\hat{\tau} = \bar{\Phi} \mathbf{A} \rho$.

**Inversion procedure**. We seek to recover the hidden object albedo $\rho$. In this case, a closed-form solution exists using the Wiener deconvolution filter and a confocal inverse filter $\mathbf{A}^{-1}$ used in non-line-of-sight imaging[20,21] (e.g., the Light-Cone Transform[22,26] or $f$–$k$ migration[23]):

$$\hat{\rho} = \mathbf{A}^{-1} \mathbf{F}^{-1} \left[ \frac{\hat{\bar{\Phi}}^*}{|\hat{\bar{\Phi}}|^2 + \frac{1}{\alpha}} \right] \mathbf{F} \tau.$$

(5)

$\mathbf{F}$ denotes the discrete Fourier transform matrix, $\hat{\bar{\Phi}}$ is the diagonal matrix whose elements correspond to the Fourier coefficients of the 3D convolution kernel, $\alpha$ is a parameter that varies depending on the signal-to-noise ratio at each frequency, and $\hat{\rho}$ is the recovered solution. Notably, the computational complexity of this method is $O(N^3 \log N)$ for an $N \times N \times N$ measurement volume, where the most costly step is taking the 3D Fast Fourier Transform. We illustrate the reconstruction procedure using CDT in Fig. 2 for a hidden scene consisting of a retroreflective letter 'S' placed ~50 cm behind the scattering layer. The initial captured 3D measurement volume is deconvolved with the diffusion model, and $f$–$k$ migration is used to recover the hidden object. A detailed description of $f$–$k$ migration and pseudocode for the inversion procedure are provided in Supplementary Notes 3–4. While the Wiener deconvolution procedure assumes that the measurements contain white Gaussian noise, we also derive and demonstrate an iterative procedure to account for Poisson noise in Supplementary Notes 5–6 (see also Supplementary Table 1 and Supplementary Figs. 7–13).

Additional captured measurements and reconstructions of objects behind the scattering medium are shown in Fig. 3. The scenes consist of retroreflective and diffusely reflecting objects: a mannequin figure, two letters at different positions (separated axially by 9 cm), and a single diffuse hidden letter. Each of these scenes is centered ~50 cm behind the scattering medium. Another captured scene consists of three traffic cones positioned behind the scattering medium at axial distances of 45, 65, and 78 cm. While retroreflective hidden objects enable imaging with shorter exposure times due to their light-efficient reflectance properties, we also demonstrate recovery of shape and position in the more general case of the diffuse letter. All measurements shown in Fig. 3 are captured by sampling a 70 cm by 70 cm grid of 32 by 32 points on the scattering layer. Exposure times and recorded photon counts for all experiments are detailed in Supplementary Table 2. The total time required to invert a measurement volume of size 32 by 32 by 128 is approximately 300 ms on a conventional CPU (Intel Core i7 9750H) or 50 ms with a GPU implementation (NVIDIA GTX 1650). We compare the reconstruction from CDT to a time-gating approach, which attempts to capture hidden object structure by isolating minimally scattered photons in a short time slice. Additional comparisons and a sensitivity analysis to the calibrated scattering parameters are described and shown in Supplementary Note 7, Supplementary Figs. 14 and 15.

**Discussion**

The approximate image formation model of Eq. (4) is valid when the difference in path length to the hidden object from two points within the illuminated spot on the far side of the scattering medium ($\mathbf{r_1} \to \mathbf{x} \to \mathbf{r_2}$) and a single illuminated point ($\mathbf{r_1} \to \mathbf{x} \to \mathbf{r_1}$) is less than the system resolution. As the standoff distance between the scattering medium and hidden object increases, this path length difference decreases, and the approximation becomes more accurate. Interestingly, speckle correlation

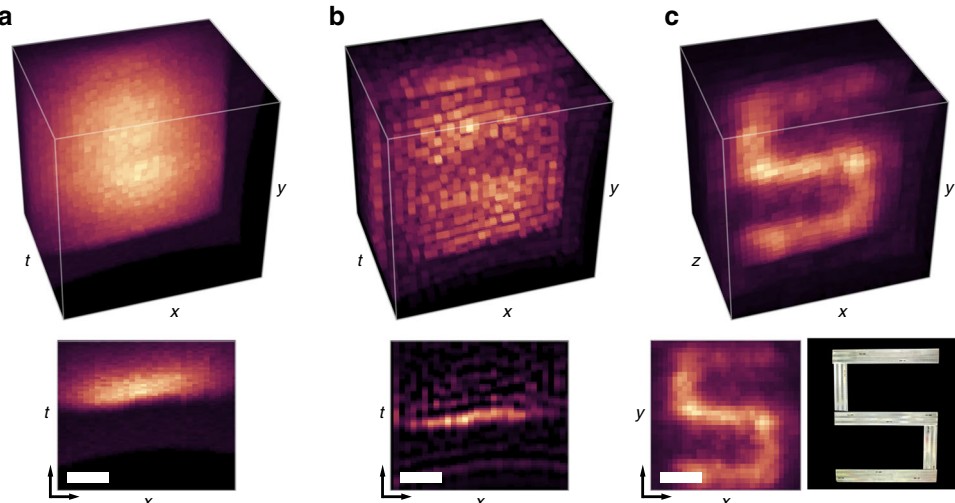

**Fig. 2 Overview of the reconstruction procedure.** The captured time-resolved measurements (**a**) are deconvolved with the calibrated diffusion operator $\bar{\Phi}$ to compensate for the time delay induced by the scattering layer and estimate a measurement volume without diffusive scattering effects (**b**, top). An $x$–$t$ slice shows the estimated streak measurement (**b**, bottom). Applying a confocal inverse filter recovers the hidden retroreflective letter 'S' (**c**, top), which resembles a photograph of the hidden scene (**c**, bottom). The reconstructed volume measures 60 cm by 60 cm by 50 cm along the x, y, and z dimensions, respectively, and a gamma of 1/3 is applied for visualization. Scalebars indicate 15 cm. The measurement volume is captured with a 1 min. acquisition time (60 ms per spatial sample). An overview of the reconstruction procedure is provided in Supplementary Movie 1.

approaches have a similar characteristic where axial range improves with standoff distance[10]. In our case, using the paraxial approximation allows us to express the condition where Eq. (4) holds as $c\Delta t > \frac{L^2}{2H}$, where $\Delta t$ is the system temporal resolution, $L$ is the lateral extent of the illuminated spot on the far side of the scattering medium, and $H$ is the standoff distance. In the diffusive regime, spreading of light causes $L$ to scale approximately as the thickness of the scattering layer, $z_d$[46]. For large incidence angles outside the paraxial regime, for example, for scanning apertures much greater than $H$, the worst-case approximation error is $\approx L$. For our prototype system, $\Delta t \approx 70$ ps and $c\Delta t \approx L$, and so even the maximum anticipated approximation error is close to the system resolution.

In practice, the imaging resolution of the system is mostly dependent on the thickness of the scattering layer and the transport mean free path, $l^* = 1/(\mu_a + \mu'_s)$. Thick scattering layers cause the illumination pulse to spread out over time, and high-frequency scene information becomes increasingly difficult to recover. The temporal spread of the pulse can be approximated using the diffusive traversal time, $\Delta t_d = \frac{z_d^2}{6Dc}$, which is the typical time it takes for a photon to diffuse one way through the medium[47]. If we take the temporal spread for two-way propagation to be approximately twice the diffusive traversal time, we can derive the axial resolution ($\Delta z$) and lateral resolution ($\Delta x$) in a similar fashion to non-line-of-sight imaging[22]. This results in $\Delta z \geq c\Delta t_d$ and $\Delta x \geq \frac{c\sqrt{w^2+H^2}}{w}\Delta t_d$, where $2w$ is the width or height of the scanned area on the scattering medium (see Supplementary Note 8, Supplementary Fig. 16). This approximation compares well with our experimental results, where $2\Delta t_d \approx 632$ ps and we measure the full width at half maximum of a pulse transmitted and back-reflected through the scattering medium to be 640 ps (shown in Supplementary Fig. 17). Thus, the axial resolution of the prototype system is ~9 cm and the lateral resolution is ~15 cm for $H = 50$ cm and $w = 35$ cm.

The current image formation model assumes that the scattering medium is of uniform thickness and composition. In practice, materials could have inhomogeneous scattering and absorption coefficients and non-uniform geometry. The proposed technique could potentially be extended to account for non-uniform scattering layer thickness by modeling the variation in geometry when performing a convolution with the solution to the diffusion equation. The confocal inverse filter could likewise be adjusted to account for scattering from nonplanar surfaces, as has been demonstrated[23]. Modeling light transport through inhomogeneous scattering media is generally more computationally expensive, but can be accomplished by solving the radiative transfer equation[48].

While we demonstrate CDT using sensitive SPAD detectors, combining CDT with other emerging detector technologies may enable imaging at still faster acquisition speeds, with thicker scattering layers, or at longer standoff distances, where the number of backscattered photons degrades significantly. For example, superconducting nanowire single-photon detectors can be designed with higher temporal resolution, lower dark-count rates, and shorter dead-times[49] than SPADs. Likewise, silicon photomultipliers (SiPMs) with photon-counting capabilities improve photon throughput by timestamping multiple returning photons from each laser pulse[50]. On the illumination side, using femtosecond lasers would potentially offer improved temporal resolution if paired with an equally fast detector, though this may require increased acquisition times if the average illumination power is decreased.

CDT is a robust, efficient technique for 3D imaging through scattering media enabled by sensitive single-photon detectors, ultra-fast illumination, and a confocal scanning system. By modeling and inverting an accurate approximation of the diffusive scattering processes, CDT overcomes fundamental limitations of traditional ballistic imaging techniques and recovers 3D shape without a priori knowledge of the target depth. We demonstrate computationally efficient reconstruction of object shape and position through thick diffusers without a priori knowledge of target position and at meter-sized scales.

## Methods

**Details of experimental setup.** In the proposed method, measurements are captured by illuminating points on the surface of the scattering media using short ($\approx$35 ps) pulses of light from a laser (NKT Katana 05HP). The pulsed laser shares an optical path through a polarizing beamsplitter (Thorlabs PBS251) with a single-pixel, single-photon avalanche diode with a $50 \times 50$ μm active area (Micro Photon Devices PDM Series Fast-Gated SPAD), which is focused on the illuminated point

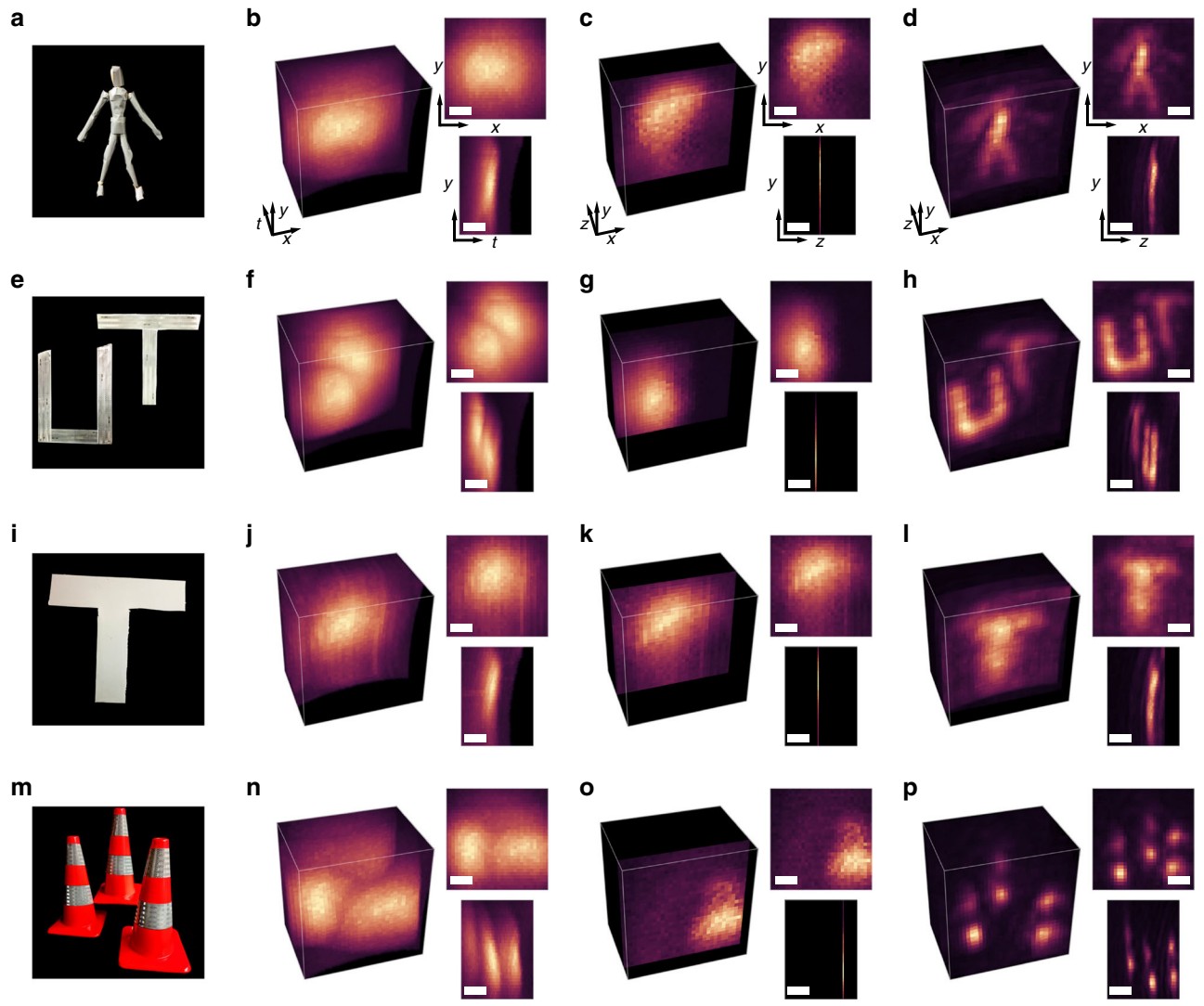

**Fig. 3 Confocal diffuse tomography reconstructions from SPAD measurements.** A photograph of the hidden retroreflective mannequin (**a**) and maximum intensity projections of the captured time-resolved measurements (**b**), a time-gated measurement slice (**c**), and the CDT reconstruction (**d**) are shown. A photograph, measurements, and reconstructions are also shown for two letter-shaped retroreflective hidden objects at different distances (**e–h**), a diffuse hidden object (**i–l**), and a group of traffic cones (**m–p**). We apply a depth-dependent scaling to the traffic cone visualization to account for radiometric falloff. Scalebars indicate 15 cm or 1 ns, and a gamma of 1/3 is applied to each maximum intensity projection[51]. Captured data are included in Supplementary Data 1 and additional visualizations are provided in Supplementary Movie 2.

using a 50-mm objective lens (Nikon Nikkor f/1.4). We use the gating capability of the SPAD to turn the detector on just before scattered photons arrive from the hidden object, and detected photons are timestamped using a time-correlated single-photon counter or TCSPC (PicoQuant PicoHarp 300). The combined timing resolution of the system is ~70 ps. The laser and SPAD are scanned onto a grid of 32 by 32 points on the surface of the scattering media using a pair of mirrors scanned with a two-axis galvanometer (Thorlabs GVS-012) and controlled using a National Instruments data acquisition device (NI-DAQ USB-6343). The pulsed laser source has a wavelength of 532 nm and is configured for a pulse repetition rate of 10 MHz with 400 mW average power. Please refer to Supplementary Fig. 1 and Supplementary Movie 1 for visualizations of the hardware prototype.

**Calibration of the scattering layer**. The reduced scattering and absorption coefficients of the scattering layer (comprising a 2.54-cm thick piece of polyurethane foam) are calibrated by illuminating the scattering layer from one side using the pulsed laser source and measuring the temporal response of the transmitted light at the other side (see Supplementary Figs. 2 and 3) using a single-pixel SPAD detector (Micro Photon Devices PDM series free-running SPAD). The captured measurement is modeled as the temporal response of the foam (given by Eq. (2)) convolved with the calibrated temporal response of the laser and SPAD. Measurements are captured for 15 different thicknesses of the scattering layer, from ~2.54 to 20.32 cm in increments of 1.27 cm. The reduced scattering and absorption coefficients are then found by minimizing the squared error between the

measurement model and the observed data across all measurements. A main source of uncertainty in the calibrated values is the value used for the extrapolation distance, which depends on the refractive index of the medium. We find that fixing the refractive index to a value of 1.12 achieves the best fit; however, to quantify uncertainty in the model parameters we also run the optimization after perturbing the refractive index within ±10% of the nominal value (from 1.01 to 1.23). The resulting parameters are $\mu'_s = 2.62 \pm 0.43$ cm$^{-1}$ and $\mu_a = 5.26 \times 10^{-3} \pm 5.5 \times 10^{-5}$ cm$^{-1}$, where the confidence intervals indicate the range containing 95% of the optimized values. We provide additional details about the optimization procedure in Supplementary Note 1.

The full width at half maximum of the spot illuminated by the laser on the far side of the scattering media is measured to be 2.2 cm, which approximately corresponds to the thickness of the scattering media (see Supplementary Fig. 5).

## Data availability
Measured data supporting the results shown in Fig. 2 and Fig. 3 are available within Supplementary Data 1. Data are also available online at https://github.com/computational-imaging/confocal-diffuse-tomography and from the authors upon request.

## Code availability
Computer code supporting the findings of this study is available within Supplementary Data 1 and online at https://github.com/computational-imaging/confocal-diffuse-tomography.

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

## Acknowledgements

D.B.L. is supported by a Stanford Graduate Fellowship in Science and Engineering. G.W. is supported by a National Science Foundation CAREER award (IIS 1553333), a Sloan Fellowship, the DARPA REVEAL program, the ARO (PECASE Award W911NF-19-1-0120), and by the KAUST Office of Sponsored Research through the Visual Computing Center CCF grant.

## Author contributions

D.B.L. conceived the method, developed the experimental setup, captured the measurements, and implemented the reconstruction procedures. G.W. supervised all aspects of the project. Both authors took part in designing the experiments and writing the paper and Supplementary Information.

## Competing interests

The authors declare no competing interests.
