## [Peer Review File · Nature Communications]

Reviewer #1 (Remarks to the Author):

Imaging through scattering media is an important and mostly open problem, which people are currently trying to attack from many different directions. In this work the authors combine time-resolved measurements with Diffused Optical Tomography to obtain a pseudo-3D reconstruction of an object hidden behind a thick scattering medium.

The technique used is interesting and, as far as I know, novel. It is also mostly explained clearly, except for a few areas that left me perplexed:

- * The Green's function for diffusion in eq 1 is only valid for an infinite medium. The moment light touches the edges, the boundary conditions change and the Green's function is deformed. In particular the tails will decay exponentially. I am not sure how much this will influence the result, but even in the best case scenario the authors should at least mention it.

- * Eq 1 assumes that D is not only constant in time, but also uniform in space. This is understandable for a proof-of-principle experiment, but this limitation (and probably a short discussion on how much more complicated things get when this assumption is violated) should be clearly discussed in the manuscript.

- * The description of the approximation made with the confocal geometry is very long and convoluted, while it can be easily summarized as $r_1 \sim r_2$. Also, eq 3 would be a lot more striking if somewhere it is shown that eq 2 has a similar form (yes, I know that eq 2 is clearly a chain of convolution products, but not everyone will be super-familiar with it).

- * One of the main mathematical tricks (the f - k migration) is mentioned at the bottom of page 8 without much context beside a reference, while it looks to me to be an important point in the implementation of the technique. While I agree that a discussion in the main text is probably too much, I would have liked some explanation about it at least in the Supplementary Information.

- * In the discussion section the "stand-off distance" is represented by the letter D . This is a very unfortunate choice, as D was already used as the diffusion coefficient, and there is a serious risk of confusion.

- * The resolution of this technique is kept carefully hidden through the whole manuscript. I understand that a complete discussion is probably best left for the Supplementary Information, but the main text must include at least a quick overview and an estimate of this value. In particular, digging in the Supplementary Information one finds that the resolution is approximately 10cm, which puts things in perspective (this is definitively not a high resolution technique).

- * The values of the absorption coefficient and the reduced scattering coefficients are presented without any error bar, but I find it difficult to believe that the uncertainty on these quantities is negligible. I would urge the authors to include those error bars.

In conclusion this is an interesting and mostly well-written work, but there are a few points that need clarification before I can recommend publication.

Reviewer #2 (Remarks to the Author):

The manuscript reports the extension of previous work by the same group on use of f - K migration methods and confocal measurements that allow to greatly simplify inverse retrieval problems that would otherwise be impossible, or close to impossible to solve. These approaches have been applied to non line of sight imaging but are here applied to imaging through scattering materials. The results are very convincing and I believe, demonstrate a significant step forward in this important research area.

The paper is very well written and clear.

I would therefore recommend publication in nature communications.

I only have one minor comment. The title and text speak of "highly scattering". Although non-descriptive adjectives are always a matter of interpretation, I feel that "highly" might be misused here.

The scattering coefficient used here is 0.07 1/cm, less than 10x smaller for example than biological tissue. And the sample is 6 scattering lengths (transport mean free paths). One could argue that with 6 scattering lengths, there is a decent component of ballistic photons - another metric by

which one might try to distinguish "scattering" from "highly scattering".

I do not want to make too much of a point about this yet, the paper does not lose any interest or impact by simply calling this "scattering" and a large part of the community will recognise themselves as working in the same regime and will therefore be very interested in this technique. I leave this with the authors. Either way, this minor point does not change in any way the quality and impact of the work.

Point-by-Point Response

We thank the reviewers for their insightful comments and appreciate their help in improving the manuscript. We have updated the paper to address their comments. In the following, we provide a point-by-point response to the raised concerns.

The paper presents a technique for non-invasive 3D imaging through scattering media at macroscopic scales based on confocal diffuse tomography. We introduce a computational imaging approach that co-designs single-photon avalanche diodes, ultra-fast pulsed lasers, and a new inverse method to facilitate new capabilities of 3D imaging through scattering media.

Referee #1 (Remarks to the Author):

- *The Green's function for diffusion in eq 1 is only valid for an infinite medium. The moment light touches the edges, the boundary conditions change and the Green's function is deformed. In particular the tails will decay exponentially. I am not sure how much this will influence the result, but even in the best case scenario the authors should at least mention it.*

We agree that the use of the Green's function of the diffusion equation should be clarified. We have adjusted the image formation model section to now state the conditions where the Green's function is valid and clarify that we are using it as an approximation. In general this is a good approximation where the thickness of the scattering media is several times greater than the transport mean free path [39].

- *Eq 1 assumes that D is not only constant in time, but also uniform in space. This is understandable for a proof-of-principle experiment, but this limitation (and probably a short discussion on how much more complicated things get when this assumption is violated) should be clearly discussed in the manuscript.*

To make the inverse scattering problem more tractable, we assume that D is constant in time and uniform in space. We now note and discuss this assumption in the Discussion section of the paper. Imaging in dynamic, anisotropic scattering media with non-uniform thickness is a challenging problem that we aim to address in future work.

- *The description of the approximation made with the confocal geometry is very long and convoluted, while it can be easily summarized as $r_1 \sim r_2$. Also, eq 3 would be a lot more striking if somewhere it is shown that eq 2 has a similar form (yes, I know that eq 2 is clearly a chain of convolution products, but not everyone will be super-familiar with it).*

This is a good point that the description of the approximation (just before Eq. 3) could be put more succinctly. We have shortened the description and also summarize the approximation as

$r_1 \sim r_2$. We also refer the readers to the Supplementary Methods, where we derive and show the full integral form of Eq. 3, which has a similar form to Eq. 2.

- *One of the main mathematical tricks (the f-k migration) is mentioned at the bottom of page 8 without much context beside a reference, while it looks to me to be an important point in the implementation of the technique. While I agree that a discussion in the main text is probably too much, I would have liked some explanation about it at least in the Supplementary Information.*

We now include a discussion of f-k migration in the supplemental information. We also provide a full implementation of the method (including f-k migration) in the Supplementary Code.

- *In the discussion section the "stand-off distance" is represented by the letter D. This is a very unfortunate choice, as D was already used as the diffusion coefficient, and there is a serious risk of confusion.*

We thank the reviewer for pointing this out, we have adjusted the symbols to avoid confusion. "H" is now used to represent the stand-off distance.

- *The resolution of this technique is kept carefully hidden through the whole manuscript. I understand that a complete discussion is probably best left for the Supplementary Information, but the main text must include at least a quick overview and an estimate of this value. In particular, digging in the Supplementary Information one finds that the resolution is approximately 10cm, which puts things in perspective (this is definitively not a high resolution technique).*

We agree that the resolution can be better described and quantified in the main text. We have augmented the Discussion section to quantify the axial resolution as approximately 10 cm and the lateral resolution as approximately 15 cm for our hardware prototype. We also include a note to refer to the Supplementary Information for further details, and now provide a table of contents for the Supplementary Information so that the resolution analysis can be located more easily.

- *The values of the absorption coefficient and the reduced scattering coefficients are presented without any error bar, but I find difficult to believe that the uncertainty on these quantity is negligible. I would urge the authors to include those error bars.*

We now provide a 95% confidence interval on the values of the absorption and scattering coefficients after analyzing the parameter covariance in the non-linear fit to the captured data. We find the absorption coefficient to be $\mu_a = 0.069 \pm 0.001 \text{ cm}^{-1}$ and the reduced scattering coefficient to be $\mu_s' = 2.41 \pm 0.02 \text{ cm}^{-1}$. The confidence intervals are now provided in the paper, and we provide additional details about the nonlinear fit and confidence bounds in the Methods.

We also demonstrate that the method is relatively robust to errors in the scattering media characterization as shown in Supplementary Figure 9.

Referee #2 (Remarks to the Author):

- *I only have one minor comment. The title and text speak of "highly scattering". Although non-descriptive adjectives are always a matter of interpretation, I feel that "highly" might be misused here.*

We thank the reviewer for this suggestion. The title has been adjusted to "Three-dimensional imaging through scattering media based on confocal diffuse tomography," and we have adjusted the wording in the abstract and text in a corresponding fashion.

Reviewer Comments, second round

Reviewer #1 (Remarks to the Author):

A few points that are still open:

* I find extremely hard to believe the uncertainties quoted by the authors for the absorption and scattering coefficients. Although it is never specified, I assume the authors are fitting their data with something like equation 13 or 14 from Applied Optics 28, 2331 (1989), which is a standard reference for these calculations. Interestingly those formulas depend exponentially on the extrapolation length, which is a very difficult number to estimate. If the extrapolation length is fixed to the wrong number, a fit will always find a best estimate for the parameters with very low variance. But this only tells us that the result is precise, not that it is accurate. I would be extremely surprised if those parameters can be estimated experimentally from a single measurement with an uncertainty smaller than 10-20%.

* Contrary to what the authors claim, the Green function they use is never a good approximation for a slab geometry. Using the same reference as above, equations 12 or 14 give the correct shape, which has exponential (and not Gaussian) tails. One might argue that the exact shape of the Green function is not important, but such a claim must be substantiated.

Please notice that these two, albeit easy to correct, are not minor points, as either could change the narrative of the paper. So both points must be addressed before the paper can be published.

Point-by-Point Response

We thank the reviewer for providing these additional comments and clarifying remarks. We have addressed each of these points below.

Reviewer #1 (Remarks to the Author):

- *I find extremely hard to believe the uncertainties quoted by the authors for the absorption and scattering coefficients. Although it is never specified, I assume the authors are fitting their data with something like equation 13 or 14 from Applied Optics 28, 2331 (1989), which is a standard reference for these calculations. Interestingly those formulas depend exponentially on the extrapolation length, which is a very difficult number to estimate. If the extrapolation length is fixed to the wrong number, a fit will always find a best estimate for the parameters with very low variance. But this only tell us that the result is precise, not that it is accurate. I would be extremely surprised if those parameters can be estimated experimentally from a single measurement with an uncertainty smaller than 10-20%.*

This is a good comment that clarifies a source of uncertainty in our calibration procedure. To provide a better estimate of the absorption and reduced scattering coefficients we recaptured the calibration data and adjusted the estimation procedure. Specifically, we use 15 time-resolved measurements of the scattering medium with thicknesses ranging from 1–8 inches (2.5–20.3 cm). Then, we optimize for the absorption and reduced scattering coefficients that provide the best fit to the data across all measurements. For the measurement model, we use a truncated version of the infinite dipole model for the slab geometry as detailed in the provided reference (Ref. 41, Applied Optics 28, 2331 (1989)).

To account for uncertainty in the extrapolation length, we calculate its value as a function of the scattering coefficient and refractive index (detailed in section 1.2 of the Supplementary Information), and we perform the optimization for a large range of fixed values for the refractive index of the material. We find that a refractive index value of 1.12 results in the best fit, but we also try a range of values within approximately 10% of this nominal value, from 1.01 to 1.23. We also optimize across a range of initializations for the scattering coefficient, as we now detail in the Methods and Supplementary Information. Overall, we perform N=253 optimizations across varying refractive index values and parameter initializations and find the average values of the reduced scattering and absorption coefficients to be $u_s' = 2.62 \pm 0.43 \text{ cm}^{-1}$ and $u_a = 0.0053 \pm$

$5.5e-5 \text{ cm}^{-1}$, where the bounds indicate the range containing 95% of the optimized parameter values (see Supplementary Figs. 2-3). For the refractive index value of 1.12 and $u_s' = 2.62 \text{ cm}^{-1}$ that produce the best overall fit, the extrapolation distance is $z_e = 3.6 \text{ mm}$.

- *Contrary to what the authors claims, the Green function they use is never a good approximation for a slab geometry. Using the same reference as above, equations 12 or 14 give the correct shape, which has exponential (and not Gaussian) tails. One might argue that the exact shape of the Green function is not important, but such claim must be substantiated.*

The reviewer raises a very good point, and we agree that a model based on the slab geometry should be used instead of the Green's function. We have therefore revised the manuscript to use the infinite dipole solution to the diffusion equation derived for the slab geometry [41, 44]. In addition to using this model to estimate the absorption and reduced scattering coefficients, we have reprocessed all experimental and simulated results using this model and updated all corresponding figures in the main manuscript and Supplementary Information.

We do not observe any remarkable qualitative differences in the experimental results as a result of updating the model. However, this is perhaps expected since both the Green's function and slab geometry models achieve a good fit to the experimental calibration data (though using different absorption and scattering coefficient values). Nevertheless, the model for the slab geometry likely results in more physically accurate calibrated parameters and is more appropriate for our experimental setup.

Reviewer Comments, third round

Reviewer #1 (Remarks to the Author):

The authors made a good job at changing their data analysis and adding enough information for the paper to be both clear and reproducible.

I have no further objection to publication.efore the paper can be published.

Point-by-Point Response

We would like to thank the reviewer for their constructive comments during the review process which were very helpful to improve the manuscript.

Reviewer #1 (Remarks to the Author):

- *The authors made a good job at changing their data analysis and adding enough information for the paper to be both clear and reproducible.
I have no further objection to publication.*